# Estimating Large Language Model Capabilities without Labeled Test Data

**Harvey Yiyun Fu    Qinyuan Ye    Albert Xu    Xiang Ren    Robin Jia**

University of Southern California, Los Angeles, CA, USA

{harveyfu, qinyuany, albertxu, xiangren, robinjia}@usc.edu

## Abstract

Large Language Models (LLMs) have the impressive ability to perform in-context learning (ICL) from only a few examples, but the success of ICL varies widely from task to task. Thus, it is important to quickly determine whether ICL is applicable to a new task, but directly evaluating ICL accuracy can be expensive in situations where test data is expensive to annotate—the exact situations where ICL is most appealing. In this paper, we propose the task of ICL accuracy estimation, in which we predict the accuracy of an LLM when doing in-context learning on a new task given only unlabeled test data for that task. To perform ICL accuracy estimation, we propose a method that trains a meta-model using LLM confidence scores as features. We compare our method to several strong accuracy estimation baselines on a new benchmark that covers 4 LLMs and 3 task collections. The meta-model improves over all baselines across 8 out of 12 settings and achieves the same estimation performance as directly evaluating on 40 collected labeled test examples per task. At the same time, no existing approach provides an accurate and reliable ICL accuracy estimation in every setting, highlighting the need for better ways to measure the uncertainty of LLM predictions.[1]

## 1 Introduction

In-context learning (ICL) with large language models (LLMs) has shown great potential in performing a wide range of language tasks (Brown et al., 2020). ICL has the unique advantages of being data-efficient (*i.e.*, only a few labeled training examples are needed) and accessible (*i.e.*, expertise in training models is no longer required). With these advantages, a non-expert user can create a system to perform a new task within minutes by writing a few examples. This gives rise to the popularity of

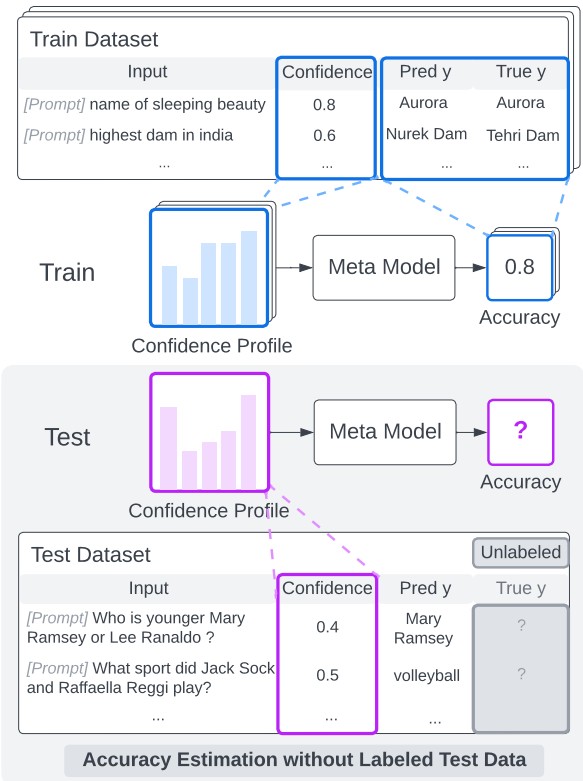

Figure 1: A demonstration of our task setting: given ICL accuracy observations of labeled training datasets, we want to estimate the dataset level ICL accuracy on the unseen test datasets without labeled test data. We propose the method of training a meta-model based on the confidence score distributions (we denote them as dataset confidence profiles)

ICL— it is being adopted and tested for a variety of use cases that stretch the boundary of what is considered possible to do with language models.

Despite the advantages of ICL, its performance is highly task-dependent (see Figure 6). It surpasses expectations on some tasks that are difficult for humans, such as answering trivia questions or riddles, but achieves near-zero performance on seemingly trivial tasks such as some text editing or spelling tasks (Srivastava et al., 2022). While evalu-

---

[1]Code is publicly released at github.com/harvey-fin/icl-estimate

ating ICL with a labeled test set is a direct solution to know whether ICL will be effective, it greatly reduces the appeal of ICL, as one of ICL's key selling points is that it does not require a large labeled dataset. In addition, many tasks do not come with a labeled test set due to high annotation costs (*e.g.*, medical/law-related questions that require professional knowledge to answer). In such cases, it is highly desirable to estimate the ICL performance without a labeled test set. This would help system developers determine whether ICL is likely to be useful for their problems of interest.

Guided by this motivation, we formalize the problem of **few-shot ICL accuracy estimation**: given a handful of labeled in-context examples and a set of *unlabeled* test examples, our goal is to estimate the overall accuracy of ICL on these test examples. Our contributions are twofold:

- We propose to address the accuracy estimation problem by training a "meta-model," which takes in LLM confidence features as input and outputs the task accuracy. The meta-model is trained with observed ICL accuracies on seen datasets, and then used to estimate ICL accuracy on unseen datasets (see Figure 1).

- We obtain 42,360 observations of LLM ICL performance, by conducting extensive ICL experiments spanning two tasks (multiple-choice QA and closed-book QA), 91 datasets, and 4 LLMs. We then benchmark the meta-model method and multiple baselines on a total of 12 evaluation settings derived from these observations.

Our meta-model can estimate ICL accuracies *without* the need for labeled test examples. In 10 out of 12 settings, the meta-model estimates are at least as accurate as directly evaluating on 16 labeled examples. In 2 out of 12 settings, they match with evaluating on 128 labeled examples. On average, we are able to save the annotation cost of 40 test labels per task by using the meta-model. Further, the meta-model outperforms all baseline methods in 8 out of 12 settings, improving the relative estimation error by 23.6% However, we also find that there exists substantial room for improvement across all settings. We envision estimating ICL accuracy without labeled test data as an open challenge and encourage the community to develop new techniques that can more accurately predict when ICL will be effective.

## 2 Related Work

### 2.1 Model Confidence and Calibration

Calibration of LLMs has been studied on a diverse range of tasks such as classification (Desai and Durrett, 2020) and question answering (QA) (Jiang et al., 2021; Kadavath et al., 2022). It aims to study whether LLMs assign meaningful correctness likelihood—also known as model confidence—to the outputs (Guo et al., 2017). Most prior work evaluates calibration at the example level (Desai and Durrett, 2020; Kamath et al., 2020); in this paper, we focus on using overall model confidence distributions to estimate dataset-level accuracies. We propose a method to learn model calibration patterns based on observations of LLMs' performance at the dataset level.

### 2.2 In-context Learning

LLMs pre-trained with auto-regressive language modeling objectives have been shown to be capable of "learning" in context when given a prompt composed of a prompt template and a few labeled demonstrations (Brown et al., 2020; Chowdhery et al., 2022). While LLMs can learn a new task only through model inference, the accuracy is sensitive to the choices of prompt templates and in-context examples (Lu et al., 2021; Zhao et al., 2021; Perez et al., 2021). Therefore, we aim to develop a method to accurately estimate ICL performance for a dataset prompted with any prompt template and combination of in-context examples.

### 2.3 Out-of-distribution (OOD) Prediction

Machine learning models in the real world commonly encounter distribution shifts between training and test time. Prior work (Guillory et al., 2021; Garg et al., 2022; Yu et al., 2022; Singhal et al., 2022; Li et al., 2022) aims to predict models' OOD performance under different setups. Garg et al. (2022) predict target domain accuracy for image classification tasks with distribution by fitting a threshold on model confidence using only labeled source data and unlabeled target data. Singhal et al. (2022) use a few additional target-domain examples to predict the accuracy, focusing on known source-target dataset pairs on which models often have low OOD accuracy due to overfitting to spurious correlations (*e.g.,* MNLI-HANS and QQP-PAWS). They find that accuracy on the given small set of target examples is a strong baseline to approximate accuracy on the full-test set. We include

the accuracy for a small set of labeled test examples as an oracle baseline (see Section 3.3). These papers all try to predict the OOD accuracy of a model trained on in-distribution training data; in contrast, in our setting we have access to some labeled datasets but the language models we study were never finetuned on those datasets. In order to avoid confusion, we instead use the terms "seen/unseen tasks" to describe the datasets available to us, rather than "in-distribution/out-of-distribution."

# 3 Accuracy Prediction

## 3.1 Problem Definition

We formalize the task of ICL accuracy estimation for unseen datasets given observations of the same model's performance on other datasets. A method for the ICL accuracy estimation task takes in four inputs: a language model $M$; a set of labeled seen datasets $\{D_i\}_{i=1}^r$, where each $D_i$ consists of a set of labeled examples $\{(x_i^{(1)}, y_i^{(1)}), \ldots, (x_i^{(n_i)}, y_i^{(n_i)})\}$ and $n_i = |D_i|$; a prompt $c$ for the test task; and an unlabeled test dataset $D_{test} = \{x_{test}^{(1)}, \ldots, x_{test}^{(m)}\}$ of size $m$. In a typical setting, each seen task should consist of a sufficient amount of labeled examples, *i.e.*, $n_i \geq 100$. The method should output the estimated accuracy of $M$ on $D_{test}$ when prompted with prompt $c$; we denote the actual accuracy of the model as $\text{acc}_{test}^{M,c}$ and $\widehat{\text{acc}}_{test}^{M,c}$ as the predicted accuracy. Note that with the labeled datasets $D_i$ and a corresponding prompt $\tilde{c}$, we can compute the corresponding dataset-level ICL accuracy $\text{acc}_i^{M,\tilde{c}}$ for $i = 1, \ldots, r$.

## 3.2 Prompt Formulation and Data Splits

We construct prompts by sampling $k$ in-context examples uniformly at random from available labeled data and formatting them with prompt templates to form a prompt (see Section B and Table 6 in the Appendix). For each dataset $D_i$, we sample $K$ prompts, each consisting of a prompt template followed by a list of in-context examples to form a prompt $c_{ij}$ for $j = 1, \ldots, K$. Note that each dataset has a training/test data split: we sample in-context examples only from the training set, and measure accuracy only on the test set. For simplicity, we use $c$ to denote a prompt in general, $C_i$ to denote the set of training prompts for dataset $D_i$, and $C_{test}$ to denote test prompts for dataset $D_{test}$.

## 3.3 Comparing with Labeled Test Data

To put our results in context, we compare all methods to the Oracle approach of sampling $l$ labeled examples from the test dataset $D_{test}$ and measuring accuracy on those $l$ examples, which we call $oracle^l$. This approach is used by Singhal et al. (2022) and it represents how well we can evaluate ICL performance for $D_{test}$ by collecting labeled examples. With a large value of $l$, we get a better evaluation of the test dataset at the cost of collecting expensive annotations. In proposing the task of accuracy prediction, we hope to develop methods that outperform the $l$-labeled oracle for values of $l$ that represent non-trivial annotation costs.

# 4 Confidence Profile Meta-Model

We propose a new method that trains a meta-model based on the *confidence profiles* of seen datasets $\{D_i\}_{i=1}^r$ to estimate ICL performance. We use the term *confidence profile* to denote the distribution of model confidence scores on each example in the dataset. We extract the confidence profiles (see Figure 1) from each seen dataset and convert them to a feature vector. We then train a meta-model to map confidence feature vectors to the dataset-level ICL accuracies. The benefits of using the confidence feature vector are twofold. First, we do not need any labeled test data, which saves annotation costs. Second, this approach is applicable to any pre-trained language model like GPT3 (Brown et al., 2020) and OPT (Zhang et al., 2022).

## 4.1 Confidence Profile

In general, given a (not-necessarily labeled) dataset $D$, LM $M$, and a prompt $c$, we obtain the confidence profile by first computing the confidence score $s^{M,c}(x)$ for each $x \in D$. The score for each input $x$ can be computed by one forward pass of $M$; the exact value of the score differs based on the task, as described below. Next, we sort the scores to obtain a list $[s_1, \ldots, s_{|D|}]$ where each $s_i \leq s_{i+1}$. Then we create a $d_{conf}$-dimensional feature vector $\text{conf}_D^{M,c}$, whose $i$-th component is a linear interpolation between $s_{\lfloor |D| \times i/d \rfloor}$ and $s_{\lceil |D| \times i/d \rceil}$. Intuitively, the $i$-th feature represents the $i/|D|$-th percentile confidence score. We refer to the feature vectors derived from confidence profiles as *confidence vectors*.

## 4.2 Confidence scores

The confidence score $s^{M,c}(x)$ is calculated differently for closed-set generation and open-ended generation.

**Closed-set generation.** Closed-set generation tasks have a pre-defined label space $\mathcal{Y}$. We take outputs from LLMs and identify the answers only by labels (Kadavath et al., 2022). For each example, we take model confidence as the normalized probability across the label space:

$$s^{M,c}(x) = \frac{p_{\hat{y}}}{\sum_{\tilde{y} \in \mathcal{Y}} p_{\tilde{y}}} \qquad (1)$$

where $p_{\tilde{y}}$ is the model-assigned probability for label $\tilde{y}$ on input $x$, $p_{\hat{y}}$ is the probability for the output label $\hat{y}$ from model $M$, and $\hat{y} = \arg\max_{\tilde{y} \in \mathcal{Y}} p_{\tilde{y}}$.

**Open-ended generation.** We refer to tasks that require sequence generation (*e.g.*, closed-book QA, summarization, machine reading comprehension, etc.) as open-ended generation tasks. We use negative log-likelihood (NLL)[2] to obtain confidence scores from each generated sequence. Let $\hat{y}$ be the model-generated sequence. We compute the confidence score as:

$$s^{M,c}(x) = -\sum_{t=1}^{|\hat{y}|} \log p_t(\hat{y}_t). \qquad (2)$$

$p_t$ is the model-assigned probability distribution at output token $t$ and $\hat{y}_t$ is the $t$-th output token

## 4.3 Meta-Model Training Data

For each seen dataset $\{D_i\}_{i=1}^r$, we sample $K$ prompts $\{c_{ij}\}_{j=1}^K$. Then for each prompt sampled we compute the confidence vector $\text{conf}_i^{M,c_{ij}}$ and accuracy $\text{acc}_i^{M,c_{ij}}$ to create one meta-training example $(\text{conf}_i^{M,c_{ij}}, \text{acc}_i^{M,c_{ij}})$. This creates a total of $r \times K$ meta-training examples. The meta-model is trained on the meta-training examples and predicts the estimated accuracy $\widehat{\text{acc}}_{test}^{M,c_{test}}$ based on the test dataset feature vector $\text{conf}_{D_{test}}^{M,c_{test}}$ for each test prompt $c_{test} \in C_{test}$. Note that since closed-set/open-ended generations have different confidence scores and accuracy evaluation metrics, the meta-model does not train on datasets that have a different task formulation than the test datasets.

---

[2]We also tried perplexity but found NLL to yield better results.

## 4.4 Meta-Model Architectures

We choose meta-models that are easy to train and contain far fewer parameters than LLMs for computational efficiency. In this paper, we consider three meta-model architectures. First, we use $k$ **Nearest Neighbors regression** ($k$-**NN**), which measures feature similarity. In the context of this paper, $k$-NN retrieves the most similar confidence profile from the seen datasets to the test dataset confidence profile and predicts based on the observed ICL accuracy on the retrieved in-distribution datasets. We use the implementation in scikit-learn library.[3] Second, we use a two-layer **Multilayer Perceptron (MLP)** that takes confidence feature vectors as input. Third, we use the tree-based method **XGBoost** (Chen and Guestrin, 2016) with the same confidence features. We use XGBoostRegressor implemented in the XGBoost library[4] and tune the hyperparameters as described in Appendix C.2.[5]

## 4.5 Evaluation

For task performance evaluation, we use Exact Match (EM) accuracy to measure accuracy for closed-set QA, and F1-score to measure accuracy for open-ended QA.

We evaluate accuracy prediction models based on absolute error, defined as $|\text{acc}_{test}^{M,c_{test}} - \widehat{\text{acc}}_{test}^{M,c_{test}}|$, where both are computed using the test dataset $D_{test}$ with a test prompt $c_{test}$. We then average the absolute error over all prompts $C_{test}$ and compute the dataset-specific mean average error:

$$err_{D_{test}} = \frac{1}{|C_{test}|} \sum_{c \in C_{test}} |\text{acc}_{test}^{M,c} - \widehat{\text{acc}}_{test}^{M,c}|.$$

Finally, to evaluate the overall success of accuracy prediction across a collection of test datasets $\mathcal{T}$, we measure mean absolute error (MAE), defined as:

$$err_{\mathcal{T}} = \frac{1}{|\mathcal{T}|} \sum_{D_{test} \in \mathcal{T}} err_{D_{test}} \qquad (3)$$

## 4.6 Baselines

We consider four baselines for accuracy estimation.

**Average training accuracy (AVGTRAIN).** We simply take the average dataset-level accuracy of

---

[3]https://scikit-learn.org/
[4]https://xgboost.readthedocs.io/en/stable/
[5]We also considered linear models but did not include them here as they perform much worse in estimating ICL accuracies.

the seen datasets as our accuracy estimation:

$$\widehat{acc}_{\text{AVGTRAIN}} = \frac{1}{r \times K} \sum_{i=1}^{r} \sum_{j=1}^{K} acc_i^{M,c_{ij}}.$$

**Average Calibration Error (AVGCONF).** We take the average confidence across the test dataset as the accuracy estimation:

$$\widehat{acc}_{\text{AVGCONF}} = \frac{1}{|C_{test}| \times m} \sum_{c \in C_{test}} \sum_{x \in D_{test}} s^{M,c}(x).$$

Note that this baseline is only applicable to closed-set generation and not open-ended generation tasks since the accuracy metric for open-ended generation (F1 score) and confidence metric (NLL) do not share the same range ($[0, 1]$ vs. $(-\infty, 0]$). The intuition behind AVGCONF is that if the model confidence scores are well-calibrated (at the example level), then the expected value of the model's confidence scores should be equal to the accuracy. In fact, we note that the MAE of AVGCONF is similar to Expected Calibration Error (ECE), which measures the example-level calibration error (Naeini et al., 2015; Guo et al., 2017; Kumar et al., 2019; Desai and Durrett, 2020).[6]

**Temperature Scaling (TS)** Temperature scaling is a widely used calibration method (Hinton et al., 2015; Guo et al., 2017; Si et al., 2022). By fitting a single scaler parameter called temperature $\tau$, it produces softer model-assigned probabilities

$$p_{\hat{y}} = \frac{\exp(z_{\hat{y}}/\tau)}{\sum_{\tilde{y}} \exp(z_{\tilde{y} \in \mathcal{Y}}/\tau)}.$$

We then obtain scaled confidence scores with Equation 1, and evaluate AVGCONF on the test dataset. Note that we optimize temperature $\tau$ based on the AVGCONF of the training datasets instead of the common approach of using NLL as an objective function.

**Average Threshold Confidence (ATC)** We use ATC (Garg et al., 2022) as one of our OOD accuracy estimation baselines. ATC takes accuracy estimation by fitting a confidence threshold on a *single* source dataset and generalizes to the target dataset. We take the estimated accuracy for the test

---

[6]ECE computes a weighted average of the difference between confidence and accuracy for each confidence interval bin. The main difference is that ECE is commonly computed by binning confidence scores into buckets, whereas in our setting each "bucket" is a different OOD dataset.

dataset to be the average of the ATC estimates from each seen dataset:

$$\widehat{acc}_{ATC} = \frac{1}{r} \sum_{i=1}^{r} atc_{i,test}^{M,c},$$

where $atc_{i,test}^{M,c}$ is the $D_i$ to $D_{test}$ ATC estimate.

### 4.7 Alternative Featurizations

In addition to the confidence profiles, we experiment with another featurization method that uses model embeddings from the LM $M$. Given a dataset $D$, LM $M$, and prompt $c$, we obtain the model embedding by first taking the last-layer, last-token embedding $e^{M,c}(x)$ for each $x \in D$, and then averaging across the dataset:

$$\widetilde{\text{embed}}_D^{M,c} = \frac{1}{|D|} \sum_{x \in D} e^{M,c}(x).$$

Since $\widetilde{\text{embed}}_D^{M,c}$ is very high-dimensional (*e.g.*, 5120 dimensional for 13B models), we use Principle Component Analysis (PCA) to reduce its dimensionality. We fit the PCA model on all dataset embedding vectors $\widetilde{\text{embed}}_D^{M,c}$ and transform them into $d_e$-dimensional vectors $\text{embed}_D^{M,c}$, which we can use as a feature vector. As an additional experiment, we can concatenate the confidence vector and embedding vector to form a combined feature vector:

$$\text{ce}_D^{M,c} = \text{conf}_D^{M,c} + \text{embed}_D^{M,c}.$$

Reducing the dimensionality makes the comparison with confidence features more fair, and does not dilute the influence of confidence features when concatenating them with embedding features.

## 5 Experiments

### 5.1 Accuracy Estimation Benchmark

We benchmark both our meta-model method for ICL accuracy estimation and the baseline methods mentioned in Section 4.6 on a total of 12 LLM-dataset collection pairs (3 dataset collections × 4 LLMs). For each evaluation setting, we evaluate 3 different featurization methods mentioned in Section 4.7. This adds up to 36 experiment settings.

**Datasets.** We use three different collections of datasets in total: multiple-choice QA (MCQA) from MMLU (Hendrycks et al., 2020) and both

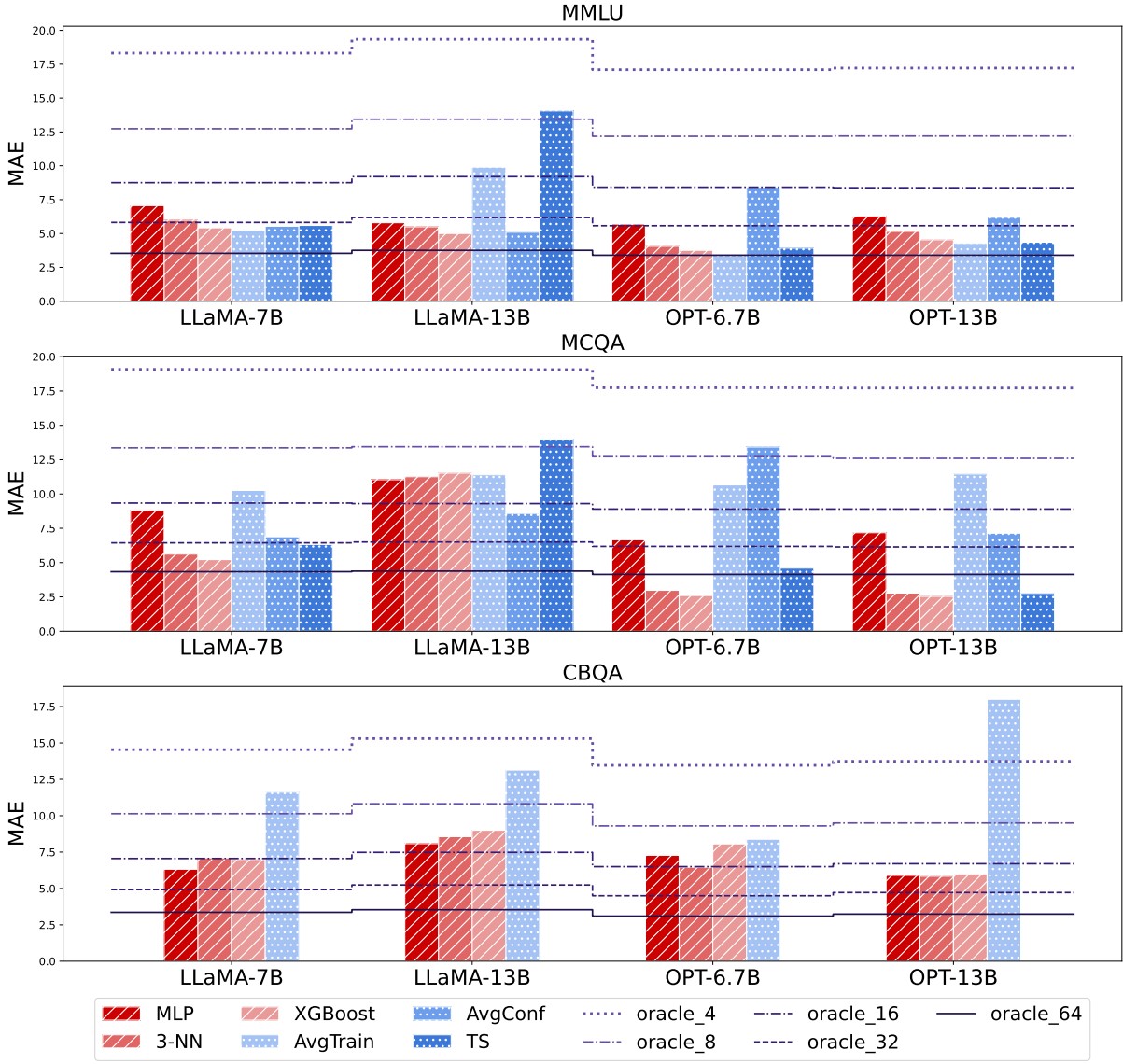

Figure 2: Bar graph of evaluation results (MAE) for all meta-models, baseline methods, and Oracle baselines of all **3** dataset collections with all **4** LLMs. We use the confidence vector as the meta-feature. Red/blue bars represent the meta-model/baseline evaluation results and the horizontal lines show the Oracle baselines.

MCQA and closed-book QA (CBQA) from Cross-Fit (Ye et al., 2021) (see Table 5 for the full list of tasks). We henceforth use MCQA and CBQA to refer to the CrossFit dataset collections respectively. We use the implementations and training/test partitions from HuggingFace Datasets (Lhoest et al., 2021). We split each collection of datasets into meta-training/test splits using 5-fold cross-validation—we partition each dataset collection into 5 equal-sized subsets and run five versions of each experiment, one with each subset being used as the meta-test set and the remaining subsets used as meta-training data. We take the average of the meta-test results as our final result.

**LLMs.** We run our experiments on four LLMs: OPT-6.7B, OPT-13B (Zhang et al., 2022), LLaMA-7B, and LLaMA-13B (Touvron et al., 2023). We use the OPT models from HuggingFace Models[7] and the LLaMA model from Meta AI.[8] More details are included in Appendix C.1.

**Experimental Details.** We generate prompts for each dataset using the method noted in Section 3.2. For each dataset in MMLU,[9] we combine the "validation" set and "dev"[10] set to be the training set

---

[7]https://huggingface.co/models
[8]https://github.com/facebookresearch/llama
[9]https://huggingface.co/datasets/cais/mmlu
[10]The "dev" set contains 5 examples for each dataset is meant for few-shot development purposes.

| Methods | LLaMA-7B | | | LLaMA-13B | | |
|---|---|---|---|---|---|---|
| | **MMLU** | **MCQA** | **CBQA** | **MMLU** | **MCQA** | **CBQA** |
| **Meta Models** | | | | | | |
| MLP | $7.06 \pm 1.03$ | $8.82 \pm 2.64$ | $\mathbf{6.32 \pm 2.13}$ | $5.80 \pm 0.63$ | $11.04 \pm 3.30$ | $\mathbf{8.08 \pm 2.99}$ |
| 3-NN | $5.98 \pm 0.97$ | $5.62 \pm 2.35$ | $7.10 \pm 2.15$ | $5.50 \pm 0.66$ | $11.26 \pm 4.56$ | $8.56 \pm 2.85$ |
| XGBoost | $5.42 \pm 4.52$ | $\mathbf{5.22 \pm 2.17}$ | $7.00 \pm 2.44$ | $\mathbf{5.00 \pm 0.62}$ | $11.52 \pm 5.20$ | $9.00 \pm 4.85$ |
| **Baselines** | | | | | | |
| AvgTrain | $\mathbf{5.26 \pm 1.40}$ | $10.26 \pm 2.50$ | $11.62 \pm 5.88$ | $9.90 \pm 1.70$ | $11.40 \pm 3.36$ | $13.14 \pm 7.25$ |
| AvgConf | $5.54 \pm 0.54$ | $6.88 \pm 3.14$ | n/a | $5.10 \pm 0.86$ | $\mathbf{8.58 \pm 1.75}$ | n/a |
| TS | $5.60 \pm 1.63$ | $6.32 \pm 3.58$ | n/a | $14.06 \pm 1.53$ | $14.00 \pm 6.95$ | n/a |
| ATC | $20.34 \pm 4.10$ | $34.66 \pm 9.36$ | $31.80 \pm 13.44$ | $20.50 \pm 4.92$ | $24.14 \pm 7.72$ | $31.80 \pm 12.49$ |
| **Oracle** | $5.82$ (**32**) | $6.44$ (**32**) | $7.06$ (**16**) | $6.18$ (**32**) | $13.44$ (**8**) | $10.82$ (**8**) |
| ACC | $31.08 \pm 6.32$ | $39.00 \pm 10.52$ | $23.7 \pm 10.44$ | $45.50 \pm 11.74$ | $50.34 \pm 12.84$ | $29.30 \pm 12.38$ |

| Methods | OPT-6.7B | | | OPT-13B | | |
|---|---|---|---|---|---|---|
| | **MMLU** | **MCQA** | **CBQA** | **MMLU** | **MCQA** | **CBQA** |
| **Meta Models** | | | | | | |
| MLP | $5.70 \pm 0.77$ | $6.66 \pm 1.25$ | $7.28 \pm 2.75$ | $6.30 \pm 1.14$ | $7.18 \pm 1.69$ | $5.90 \pm 1.18$ |
| 3-NN | $4.06 \pm 0.33$ | $2.98 \pm 0.57$ | $\mathbf{6.46 \pm 2.05}$ | $5.16 \pm 0.24$ | $2.78 \pm 1.06$ | $\mathbf{5.84 \pm 2.19}$ |
| XGBoost | $3.76 \pm 0.32$ | $\mathbf{2.60 \pm 0.73}$ | $8.06 \pm 3.07$ | $4.54 \pm 0.35$ | $\mathbf{2.54 \pm 1.16}$ | $6.00 \pm 2.43$ |
| **Baselines** | | | | | | |
| AvgTrain | $\mathbf{3.48 \pm 0.37}$ | $10.66 \pm 1.75$ | $8.38 \pm 3.86$ | $\mathbf{4.28 \pm 0.35}$ | $11.46 \pm 2.15$ | $18.00 \pm 3.77$ |
| AvgConf | $8.42 \pm 0.60$ | $13.42 \pm 1.70$ | n/a | $6.20 \pm 0.77$ | $7.14 \pm 1.77$ | n/a |
| TS | $3.92 \pm 0.17$ | $4.60 \pm 0.80$ | n/a | $4.36 \pm 0.31$ | $2.72 \pm 1.03$ | n/a |
| ATC | $24.54 \pm 2.12$ | $37.64 \pm 7.93$ | $32.16 \pm 14.60$ | $24.72 \pm 3.32$ | $29.40 \pm 10.28$ | $30.34 \pm 12.48$ |
| **Oracle** | $5.58$ (**32**) | $2.76$ (**128**) | $6.50$ (**16**) | $5.58$ (**32**) | $2.76$ (**128**) | $6.70$ (**16**) |
| ACC | $26.68 \pm 4.42$ | $33.16 \pm 9.80$ | $17.54 \pm 5.92$ | $26.82 \pm 5.28$ | $33.68 \pm 12.06$ | $19.34 \pm 6.30$ |

Table 1: Evalutaion results (MAE) and variations (SD) for all **4** LLMs, **3** dataset collections settings, using confidence vector as the meta-feature. For the Oracle baselines, we include the closest lower-bound. *e.g.*, if the error is between $oracle^{32}$ and $oracle^{64}$, we put $oracle^{32}$ (32). OPT-13B settings have the lowest average MAE (5.13), compared to OPT-6.7B (5.28), LLaMA-7B (6.50), LLaMA-13B(8.41). We report overall accuracy (ACC) by Exact Match for MMLU and MCQA, and F1-score for CBQA

that we sample in-context examples from. We sample **10** 3-shot, **10** 4-shot, and **10** 5-shot prompts[11] and decorate each of them with 5 prompt templates chosen for MMLU (see Table 3). We choose $d_{conf} = 20$ here since many of the datasets contain only 100 text examples. For MCQA and CBQA, we sample in-context examples from a pool of 100 examples as the training set, and obtain a test set of 1000 examples (see Section A in the appendix for implementation details). For each dataset, we sample **30** 3-shot and **30** 4-shot prompts and decorate them with only the null template. We choose $d_{conf} = 100$ for both MCQA and CBQA settings. Section 5.3 contains more details about the ablation studies for $d_{conf}$. Due to computational reasons, we sample only 30 prompts (compared to 60

prompts for MCQA/CBQA datasets) for MMLU because it contains a very large number of datasets.

## 5.2 Main Results

**The meta-model outperforms all baselines under certain evaluation settings.** Table 1 shows the meta-model estimation error for each evaluation setting. For 8 out of 12 settings (all CBQA settings, LLaMA-7B on MCQA, LLaMA-13B on MMLU, both OPT models on MCQA), the best meta-model architecture has 23.67% lower relative MAE than the best baseline method on average. In the best case (OPT-6.7B on MCQA), the meta-model can achieve 43.5% lower relative MAE than all baselines. However, for the other 4 settings (both OPT models on MMLU, LLaMA-7B on MMLU, and LLaMA-13B on MCQA), baseline methods provide more accurate estimates of ICL accuracy. Fig-

---

[11] We choose up to 5-shot setting because it is studied in previous studies (Touvron et al., 2023; Rae et al., 2021).

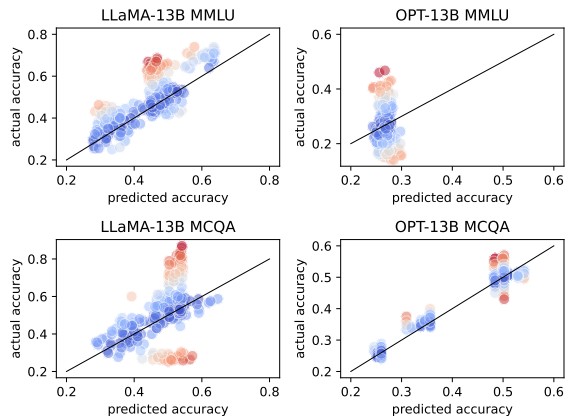

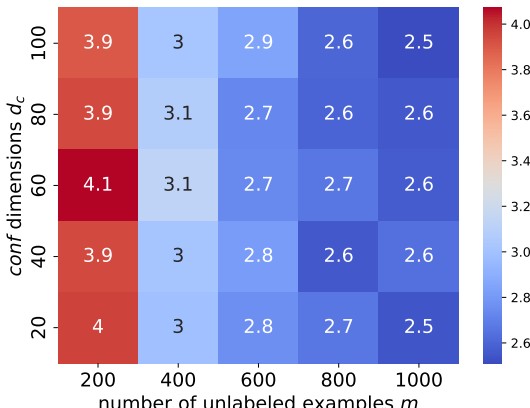

Figure 3: We plot the meta-model predicted accuracy versus the actual accuracy across 4 settings. We use the XGBoost meta-model and the confidence vector meta-feature. Each entity represents an observation for one dataset. Red/blue represents higher/lower absolute error.

Figure 4: Estimation results for ablating the number of unlabeled examples $m$ (x-axis) and confidence vector dimensions $d_c$ (y-axis), evaluated on the OPT-13B on MCQA using the XGBoost meta-model.

ure 2 shows the evaluation results graphically. On average across all 12 settings, the best estimation errors from the meta-models are 32.5% less than the actual accuracy standard deviations. In 11 out of 12 settings, the estimation errors are within one standard deviation of the actual accuracy.

**Oracle baselines indicate useful accuracy estimations.** In comparison to the Oracle baselines, the meta-model outperforms the $oracle^{32}$ baseline in all MMLU and MCQA settings except for LLaMA-13B on MCQA (achieves $oracle^{8}$) and outperforms the $oracle^{16}$ baseline in all CBQA settings except for LLaMA-13B (achieves $oracle^{8}$). In the two best-case settings (using XGBoost as the meta-model on MCQA with either OPT model), the meta-model achieves the $oracle^{128}$ baseline, i.e., is equivalent to estimating the accuracy using 128 annotations.

**Baseline methods are effective in some settings.** While ATC is a weak baseline for ICL accuracy estimation, AVGTRAIN, AVGCONF, and TS are strong baselines for MMLU and MCQA. AVG-TRAIN is able to achieve $oracle^{32}$ for 3 out of 12 settings (LLaMA-7B, OPT-6.7B, and OPT-13B on MMLU); AVGCONF is able to achieve $oracle^{32}$ for 2 settings (LLaMA-7B and LLaMA-13B on MMLU), and $oracle^{16}$ for 3 settings (both LLaMA models on MCQA and OPT-13B on MMLU). Note that temperature scaling improves calibration in certain settings: TS is able to achieve $oracle^{128}$ for one setting (OPT-13B on MCQA), and $oracle^{32}$

for 5 settings (LLaMA-13B on MMLU, both OPT models on MMLU, and OPT-6.7B on MCQA).[12]

**Ablation on Model Architecture** Across three meta-model structures, the XGBoost meta-model overall provides the most accurate estimation as it has the lowest MAE for 7 out of 12 evaluation settings. The average MAE is 5.88 for XG-Boost meta-models, 5.94 for 3-NN meta-models, and 7.18 for MLP meta-models. Surprisingly, 3-NN meta-models have a lower average MAE than MLP meta-models despite having a simpler model structure. In Figure 3, we show that the XGBoost meta-model provides well-correlated accuracy estimation across 4 different evaluation settings.

**Ablation on Featurization Methods** We consider three featurization methods as described in Section 4.7. Table 2 in the appendix shows that the best overall accuracy estimation for all settings is attained by using the confidence vectors as meta-features (achieves the lowest MAE for 26 out of 36 evaluation settings). The average MAE is 6.27 for $conf$, 8.34 for $embed$, and 7.36 for $ce$. Further, using $conf$ as features demonstrates a more dominant advantage for all CBQA tasks, achieving the lowest MAE for 11 out of 12 evaluation settings.

### 5.3 Effect of Unlabeled Data and Confidence Vector Dimensions

We now study confidence feature vector ablations by varying the number of unlabeled test examples $m$ in each unseen dataset and the dimension of the

---

[12]As noted in Section 4.6, the AVGCONF and TS baseline is not applicable to CBQA datasets.

confidence vector $d_{conf}$. We test with OPT-13B on MCQA datasets using the XGBoost meta-model since we achieve the lowest MAE in this setting. Figure 4 shows that increasing $m$ enables better accuracy estimation, reducing the average MAE (across all $d_{conf}$) from 3.92 for $m = 200$ to 2.56 for $m = 1000$. Note that increasing $m$ requires performing additional LLM inferences on unlabeled examples, so leveraging unlabeled test data is constrained by computational cost considerations. The quality of our accuracy estimates does not vary much as we change the confidence vector dimension $d_{conf}$, as shown in Figure 4.

### 5.4 Effect of Number of Shots

We compare ICL accuracy estimation performance given different $k$-shot ICL accuracy observations for LLaMA-13B on MMLU datasets. Table 4 in the Appendix shows that the meta-model produces a slightly better ICL accuracy estimation for the 3-shot setting. Overall, the meta-model gives consistent accuracy estimates across different $k$-shot settings as they all achieve $oracle^{32}$.

### 5.5 Prompt Selection

Previous works demonstrated that ICL performance is highly sensitive to the prompt templates as well as in-context examples (Zhao et al., 2021; Perez et al., 2021; Chen et al., 2022); we are thus interested in whether our ICL accuracy estimation method can be applied to select the best ICL prompt $c \in C_{test}$ for the test dataset. For each dataset, we use the XGBoost meta-model to select the best prompt $\widehat{c^*}$, as opposed to the actual best prompt $c^*$. We then compute the corresponding ICL accuracies and compare them to the average accuracy across all test prompts. Figure 5 shows that there is a significant difference in ICL accuracy given different prompts for all 12 settings, and the selected prompts lead to better ICL accuracies than the average accuracy for 7 out of 12 settings. On average, the selected prompt is 15.6% as effective as the actual best prompt. The limited improvement from the random baseline indicates there's a large room for improvement and we encourage future work to derive a better prompt selection standard.

### 6 Discussion and Conclusion

In this paper, we study the problem of few-shot ICL accuracy estimation. We propose training a meta-model based on LLM confidence features and

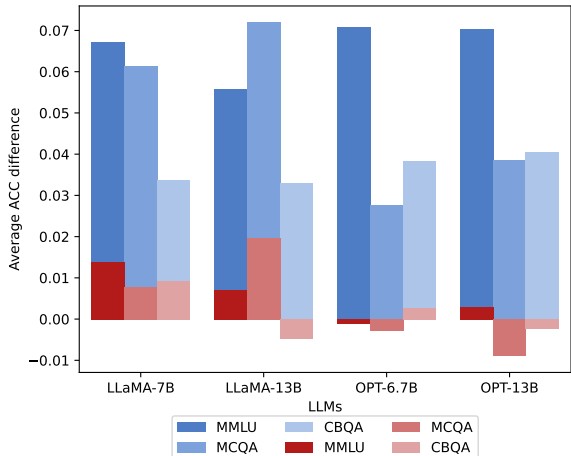

Figure 5: Prompt selection results for all evaluation settings, measured by the absolute difference between ICL accuracy when prompted with $c$ and the average accuracy. Blue bars show the actual best prompt (*i.e.*, $c = c^*$), and red bars show the selected best prompt (*i.e.*, $c = \widehat{c^*}$)

observed accuracies on seen datasets. We show that without using any labeled test data, the meta-model is often able to attain accurate estimates of ICL accuracy, which is practically useful for predicting LLMs' accuracy on datasets that have high annotation costs. We also construct a large-scale benchmark for dataset-level ICL accuracy estimation by evaluating the meta-model and multiple baseline methods across 12 evaluation settings and 3 meta-feature options. We observe that while some baseline methods can provide good accuracy estimates, our meta-model demonstrates non-trivial improvement in estimation abilities over baseline methods in 8 out of 12 evalutaion settings.

We encourage future work to develop better meta-model architectures as well as better meta-features and study potential implications for the meta-model, such as acting as a prompt template/ICL example selection method. We believe that our benchmark can serve as an open challenge for improving dataset-level ICL accuracy estimations, leading to an improved understanding of when ICL is likely to be effective.

### Limitations

While we conducted extensive experiments to study ICL accuracy estimations, there are many more LLMs that have exhibited impressive capabilities on a variety of tasks. Due to computational constraints, we do not benchmark accuracy estimations based on LLMs with limited access (*e.g.*,

GPT-4 (OpenAI, 2023)) as it is difficult to extract model embedding features, or those larger than 13B. We also don't consider instruction-tuned models to avoid possible overlaps between their training datasets and our evaluation datasets. Meanwhile, instruction tuning sometimes hurts model performance on canonical datasets such as MMLU, as shown in Gudibande et al. (2023). It might also significantly hurt calibration as reported in OpenAI (2023). For the same reasons, we include only a limited number of prompt templates and in-context example variations for ICL prompting. While we choose only 3 few-shot settings for MMLU and 2 for MCQA and CBQA, it is possible to achieve better accuracy estimations with more observations in the training data.

In terms of dataset selection, we use 13 closed-book QA tasks for the open-ended generation setting. Our findings might not generalize to other open-ended generation tasks such as summarization or long-form question answering. Overall, the meta-model provides effective accuracy estimations, but there's still substantial room for improvement.

## Acknowledgements

We would like to thank Ting-Yun Chang for her valuable contributions. We thank Ameya Godbole, Johnny Wei, and Wang Zhu for their valuable discussions. We also thank all members of the Allegro lab at USC for their support and valuable feedback. RJ was supported by an Open Philanthropy research grant, a Cisco Research Award, and HF was supported by the USC Provost Fellowship Award.

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

| MMLU | | | | | | | | | | | | |
|---|---|---|---|---|---|---|---|---|---|---|---|---|
| **Methods** | **LLaMA-7B** | | | **LLaMA-13B** | | | **OPT-6.7B** | | | **OPT-13B** | | |
| **Meta Models** | conf | embed | ce | conf | embed | ce | conf | embed | ce | conf | embed | ce |
| MLP | 7.06 | 9.76 | 8.00 | 5.80 | 9.92 | 8.84 | 5.70 | 10.66 | 8.02 | 6.30 | 11.24 | 9.48 |
| 3-NN | 5.98 | 5.38 | 5.38 | 5.50 | 6.80 | 6.80 | 4.06 | 4.60 | 4.60 | 5.16 | 5.20 | 5.20 |
| XGBoost | 5.42 | 4.52 | **4.56** | **5.00** | 7.30 | 5.16 | 3.76 | 3.94 | 3.76 | 4.54 | 4.88 | 4.72 |
| **Baselines** | | | | | | | | | | | | |
| AVGTRAIN | 5.26 ± 1.40 | | | 9.90 ± 1.70 | | | **3.48 ± 0.37** | | | **4.28 ± 0.35** | | |
| AVGCONF | 5.54 ± 0.54 | | | 5.10 ± 0.86 | | | 8.42 ± 0.60 | | | 6.20 ± 0.77 | | |
| TS | 5.60 ± 1.63 | | | 14.06 ± 1.53 | | | 3.92 ± 0.17 | | | 4.36 ± 0.31 | | |
| ATC | 20.34 ± 4.10 | | | 20.50 ± 4.92 | | | 24.54 ± 2.12 | | | 24.72 ± 3.32 | | |
| **Oracle** | 5.82 (**32**) | | | 6.18 (**32**) | | | 5.58 (**32**) | | | 5.58 (**32**) | | |
| ACC | 31.08 ± 6.32 | | | 45.50 ± 11.74 | | | 26.68 ± 4.42 | | | 26.82 ± 5.28 | | |
| **MCQA** | | | | | | | | | | | | |
| **Methods** | **LLaMA-7B** | | | **LLaMA-13B** | | | **OPT-6.7B** | | | **OPT-13B** | | |
| **Meta Models** | conf | embed | ce | conf | embed | ce | conf | embed | ce | conf | embed | ce |
| MLP | 8.82 | 12.50 | 10.62 | 11.04 | 16.86 | 14.64 | 6.66 | 13.00 | 10.30 | 7.18 | 14.54 | 13.72 |
| 3-NN | 5.62 | 6.22 | 6.22 | 11.26 | 11.24 | 11.24 | 2.98 | 3.84 | 3.84 | 2.78 | 5.60 | 5.58 |
| XGBoost | **5.22** | 6.34 | 6.12 | 11.52 | 10.90 | 11.34 | **2.60** | 5.34 | 3.00 | **2.54** | 6.74 | 2.94 |
| **Baselines** | | | | | | | | | | | | |
| AVGTRAIN | 10.26 ± 2.50 | | | 11.40 ± 3.36 | | | 10.66 ± 1.75 | | | 11.46 ± 2.15 | | |
| AVGCONF | **6.88 ± 3.14** | | | **8.58 ± 1.75** | | | 13.42 ± 1.70 | | | 7.14 ± 1.77 | | |
| TS | 6.32 ± 3.58 | | | 14.00 ± 6.95 | | | 4.60 ± 0.80 | | | 2.72 ± 1.03 | | |
| ATC | 34.66 ± 9.36 | | | 24.14 ± 7.72 | | | 37.64 ± 7.93 | | | 29.40 ± 10.28 | | |
| **Oracle** | 6.44 (**32**) | | | 9.30 (**16**) | | | 2.76 (**128**) | | | 2.76 (**128**) | | |
| ACC | 39.00 ± 10.52 | | | 50.54 ± 13.86 | | | 33.42 ± 11.58 | | | 33.68 ± 12.06 | | |
| **CBQA** | | | | | | | | | | | | |
| **Methods** | **LLaMA-7B** | | | **LLaMA-13B** | | | **OPT-6.7B** | | | **OPT-13B** | | |
| **Meta Models** | conf | embed | ce | conf | embed | ce | conf | embed | ce | conf | embed | ce |
| MLP | **6.32** | 14.26 | 7.14 | **8.08** | 12.18 | 10.96 | 7.28 | 8.44 | 8.46 | 5.90 | 7.42 | 9.12 |
| 3-NN | 7.10 | 10.74 | 9.14 | 8.56 | 10.74 | 10.72 | **6.46** | 10.44 | 10.18 | **5.84** | 10.62 | 10.64 |
| XGBoost | 7.00 | 13.10 | 8.56 | 9.00 | 11.60 | 10.64 | 8.06 | 7.58 | 7.74 | 6.00 | 8.54 | 7.74 |
| **Baselines** | | | | | | | | | | | | |
| AVGTRAIN | 11.62 ± 5.88 | | | 13.14 ± 7.25 | | | 8.38 ± 3.86 | | | 18.00 ± 3.77 | | |
| ATC | 31.80 ± 13.44 | | | 31.80 ± 12.49 | | | 32.16 ± 14.60 | | | 30.34 ± 12.48 | | |
| **Oracle** | 7.06 (**16**) | | | 10.82 (**8**) | | | 6.50 (**16**) | | | 6.70 (**16**) | | |
| ACC | 23.72 ± 10.44 | | | 29.30 ± 12.38 | | | 17.54 ± 5.92 | | | 19.34 ± 6.30 | | |

Table 2: Full estimation results for all **4** LLMs, **3** dataset collections, and **3** meta-feature choices. XGBoost is the best overall meta-model structure with an average MAE of 6.60. The confidence vector is the best overall feature with an average MAE of 6.38 across all evaluation settings. The best meta-model outperforms all baseline methods in 9 out of 12 evaluation settings. We report overall accuracy (ACC) by Exact Match for MMLU and MCQA, and F1-score for CBQA

# A  Dataset Implementation Details

Following Section 5.1's discussion of specific dataset implementations for MCQA and CBQA, our approach for each setting is: for tasks that have a defined train/test split in HuggingFace Datasets, we sample in-context examples from the first 100 examples in the training set and then choose the first 1000 examples from the test set to be our test questions; while for other datasets that only have the training set defined, we first sample a subset of 100 examples from the first 80% of the dataset and then sample in-context examples from the subset. The test questions are obtained by sampling 1000 examples from the rest 20% of the dataset.

# B  Prompt Templates

We collect 5 general prompt templates for MMLU datasets: 1 Null template, 1 self-constructed prompt template, 2 from previous work (Rae et al., 2021; Hendrycks et al., 2020), and 1 generated by ChatGPT. For MCQA and CBQA, we only use the Null template due to resource considerations.

| Prompt Template | Source |
|---|---|
| Null | n/a |
| You are an expert in **{subject}**, here are some multiple-choice questions | Self-constructed |
| The following are multiple choice questions (with answers) about **{subject}** | Hendrycks et al. (2020) |
| A highly knowledgeable and intelligent AI answers multiple-choice questions about **{subject}** | Rae et al. (2021) |
| Test your knowledge of **{subject}** with multiple-choice questions and discover the correct answers. | ChatGPT |

Table 3: Prompt templates for MMLU datasets. **{subject}** is replaced by the dataset subsets (*e.g.*, abstract_algebra)

# C  Implementation Details

We will release code to reproduce our results upon publication.

## C.1  LLMs Implementations

We use LLaMA-7B and LLaMA-13B (Touvron et al., 2023) from Meta AI, and OPT-6.7B and OPT-13B (Zhang et al., 2022) from HuggingFace transformers. For LLaMA-7B, OPT-6.7B, and OPT-13B, we run evaluations using a single RTXA6000 GPU (48GB). For LLaMA-13B, we run evaluations by parallel inference on two RTXA6000 GPUs. We use half-precision for both OPT models. Note that some LLMs (except for OPT-13B) can be run on GPUs with a smaller memory. We evaluate 42,360 ICL observations in total, where each observation is a dataset with 100 to 1000 examples. The total inference process takes around 2000 GPU hours.

## C.2  Meta-model Implementations

All meta-model architectures can be trained on an i7-10700 CPU. The total training time of the meta-model on one experiment setting varies from 1.5 hours to 24 hours depending on training data dimensions. We include the implementation details for each of the meta-model architectures. We use the random seed 1 for all processes involving randomness. For **K Nearest Neighbors regression**, we use the implementation of KNeighborsRegressor from sklearn library. We use euclidean distance as the weight metric, and fit the model on the meta-training data.

For **MLP**, we implement with Pytorch. We use a 2-layer MLP with the size of the $hidden\_state = 1536$, $learning\_rate = 1e - 5$, and $dropout\_rate = 0.2$. We use Adam Optimizer and MSELoss, and we perform early stopping with the validation data. The validation data is a 20% random partition of the meta-training data. For early stopping, the max epoch is 50 and the patience is 7.

For **XGBoost**, we implement with the XGBRegressor from the XGBoost library(Chen and Guestrin, 2016). We use a 5-fold random search cross-validation for 300 iterations to choose the hyperparameter. The candidates are:

```
{
    "lr": uniform(0.01, 0.5),
    "max_depth": randint(3,10),
    "n_estimators": randint(100, 1000),
    "colsample_bytree": [0.5, 0.6, 0.7, 0.8, 0.9, 0.
        95, 1],
    "subsample": [0.5, 0.6, 0.7, 0.8, 0.9, 0.95, 1],
    "gamma": uniform(0,1),
    "reg_alpha": uniform(0,1),
    "reg_lambda": uniform(0,1)
}
```

## C.3  Other Implementation Details

**Output Collection**  For multiple-choice QA tasks (MMLU and MCQA), we collect generated choice labels (e.g., "(A)") from the first 5 tokens generated. For closed-book QA tasks (CBQA), we collect the first 16 newly generated tokens as the model output and truncate the outputs by the *newline* token.

**Temperature Scaling**  We search for the optimal temperature $\tau$ based on the meta-training set. The search grid is:

```
np.linspace(1.0, 3.0, 100)
```

## D  Few-shot setting ablation results

| Methods | 3-shot | 4-shot | 5-shot | mixed |
|---|---|---|---|---|
| **Meta Models** | | | | |
| | **conf** | **conf** | **conf** | **conf** |
| MLP | 5.54 | 5.92 | 5.90 | 5.80 |
| 3-NN | 5.34 | 5.80 | 5.92 | 5.50 |
| XGBoost | 4.86 | 5.34 | 5.40 | 5.00 |
| **Baselines** | | | | |
| Avg | 9.78 | 9.92 | 10.16 | 9.90 |
| ACE | 5.00 | 5.10 | 5.18 | 5.10 |
| ATC | 20.10 | 20.44 | 20.68 | 20.50 |
| **Oracle** | 6.14 (**32**) | 6.14 (**32**) | 6.12 (**32**) | 6.18 (**32**) |
| ACC | $45.50_{\pm 11.44}$ | $45.36_{\pm 11.26}$ | $45.52_{\pm 12.00}$ | $45.50_{\pm 11.74}$ |

Table 4: Estimation results for separate and mixed few-shot settings, measured by MAE and tested on the LLaMA-13B on the MMLU setting. The accuracy estimation is consistent across different shot settings. We report the accuracy (ACC) as Exact Match accuracy

| Dataset Collection | Datasets |
|---|---|
| **MMLU** | "abstract_algebra", "anatomy", "astronomy", "college_biology"
"college_chemistry", "college_computer_science", "college_mathematics", "college_physics"
"computer_security", "conceptual_physics", "electrical_engineering", "elementary_mathematics"
"high_school_biology", "high_school_chemistry", "high_school_computer_science"
"high_school_mathematics", "high_school_physics", "high_school_statistics"
"machine_learning", "high_school_government_and_politics" , "high_school_geography"
"econometrics", "high_school_macroeconomics", "high_school_microeconomics", "sociology"
"high_school_psychology", "human_sexuality", "professional_psychology", "public_relations"
"security_studies", "us_foreign_policy", "formal_logic", "high_school_european_history"
"high_school_us_history", "high_school_world_history", "international_law", "jurisprudence"
"logical_fallacies", "moral_scenarios", "philosophy", "prehistory", "professional_law"
"world_religions", "business_ethics", "clinical_knowledge", "college_medicine", "global_facts"
"human_aging", "management", "marketing", "medical_genetics", "miscellaneous", "nutrition"
"moral_disputes", "professional_accounting", "professional_medicine", "virology" |
| **MCQA** | "wiqa", "wino_grande", "swag", "superglue-copa", "social_i_qa"
"race-middle", "quartz-with_knowledge", "quartz-no_knowledge", "quarel", "qasc"
"openbookqa", "hellaswag", "dream", "cosmos_qa", "commonsense_qa"
"ai2_arc", "codah", "aqua_rat", "race-high", "quail", "definite_pronoun_resolution" |
| **CBQA** | "squad-no_context", "numer_sense", "kilt_trex", "kilt_zsre", "lama-trex"
"lama-squad", "lama-google_re", "lama-conceptnet", "kilt_hotpotqa"
"kilt_nq", "freebase_qa", "web_questions", "jeopardy" |

Table 5: We list 3 dataset collections: MMLU, MCQA, and CBQA. There are 57 datasets in MMLU, 21 datasets in MCQA, and 13 datasets in CBQA.

| Task | Examples | Labels |
|------|----------|--------|
| MMLU | What is the second most common element in the solar system?
(A) Iron
(B) Hydrogen
(C) Methane
(D) Helium

Answer: (D) Helium

On which planet in our solar system can you find the Great Red Spot?
(A) Venus
(B) Mars
(C) Jupiter
(D) Saturn

Answer: | 













(C) |
| CBQA | Who are the members of Aespa?

Answer: Karina, Giselle, Winter, and Ningning

Who is the first overall pick in the 2011 NBA draft?

Answer: | 




Kyrie Irving |

Table 6: Two 1-shot examples decorated with a default null prompt template for MMLU datasets and CBQA dataset. We take the same prompt template for MCQA as MMLU.

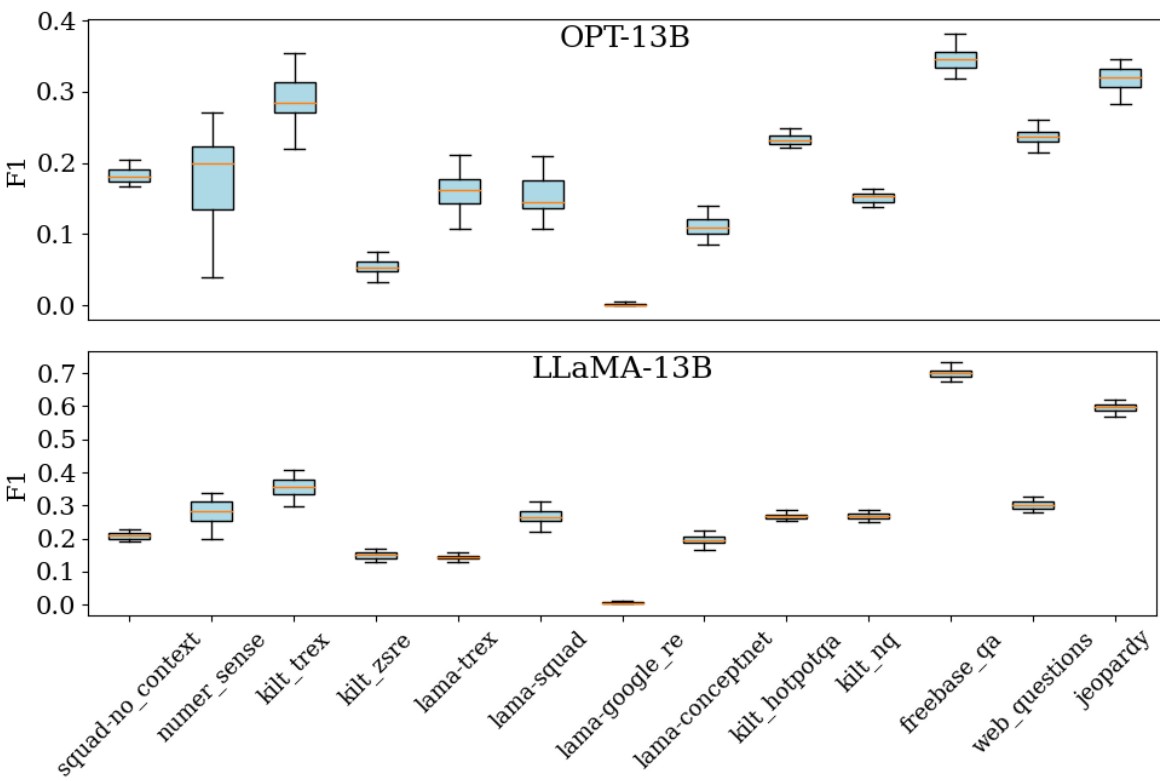

Figure 6: 4-shot ICL accuracy for OPT-13B and LLaMA-13B on CBQA tasks, where each boxplot summarizes the F1 scores over 30 ICL prompt variations of one dataset. Both OPT-13B and LLaMA-13B have large variances across different tasks, showing the challenge of ICL accuracy estimation.