# OpenReview forum: "Estimating Large Language Model Capabilities without Labeled Test Data"
_EMNLP/2023/Conference — EMNLP 2023 Findings_

### Official Review · Reviewer_cyhj · 2023-07-29

**Typos Grammar Style And Presentation Improvements:** 1. May be the '91' (Line 78) is a typo.
**Soundness:** 3

**Excitement:**

3: Ambivalent: It has merits (e.g., it reports state-of-the-art results, the idea is nice), but there are key weaknesses (e.g., it describes incremental work), and it can significantly benefit from another round of revision. However, I won't object to accepting it if my co-reviewers champion it.

**Paper Topic And Main Contributions:**

This paper focuses on estimating the in-context learning (ICL) ability of LLMs on different tasks. Two main contributions are: 1. first formalizing the problem of few-shot ICL accuracy estimation: previous works focus on example-level calibration, but this work proposes to work on dataset-level estimation. (new problem, new benchmark) 2. proposing the method of training meta-model based on LLM confidence and accuracy for the dataset-level estimation problem. (new method)

**Questions For The Authors:**

1. Line 78 'conducting extensive ICL experiments spanning 91 datasets', I didn't see the results on 91 datasets in the subsequent sections.
2. It seems that the problem is highly influenced by label distribution of a task?


**Reasons To Accept:**

1. The problem proposed in this paper is interesting, and the authors formalize the problem well, accompanied by several strong baselines.
2. The confidence profile meta-model is effective across different datasets on the dataset-level ICL estimation and shows potential in prompt selection.

**Reasons To Reject:**

1. As a general problem for ICL, the benchmark proposed in this paper should include more tasks and organize them better.
2. Table 1 is hard to read, maybe a better metric for this problem is required.
3. The estimation variance is also critical for this problem, how is the variance of each method (including baselines) under different prompt and settings, does the meta-model shows a small variance across various scenarios? If the authors can elaborate on this point, the method's effectiveness will be further confirmed.

**Reproducibility:**

4: Could mostly reproduce the results, but there may be some variation because of sample variance or minor variations in their interpretation of the protocol or method.

**Reviewer Confidence:**

2: Willing to defend my evaluation, but it is fairly likely that I missed some details, didn't understand some central points, or can't be sure about the novelty of the work.

---

> ### Author Rebuttal · Authors · 2023-08-29
>
> We thank the reviewer for the insightful questions and comments! Here are our responses to the individual questions:
>
> > **Q1:** As a general problem for ICL, the benchmark proposed in this paper should include more tasks and organize them better.
>
> We agree with the reviewer that including more tasks would add to the comprehensiveness. Considering the number of permutations of prompts and in-context examples, we actually evaluate a total of **42,360** in-context learning performance observations based on **91** datasets and **4** LLMs. While we would be interested in running experiments on additional datasets, the number of datasets we can evaluate is subject to computational resource limitations and we motivate future work to include evaluations on more tasks.
>
> > **Q2:** Table 1 is hard to read, maybe a better metric for this problem is required.
>
> We are sorry for the inconvenience. The oracle baselines and the estimation errors might not be intuitive. We propose to include graphical representations of the oracle baselines from $oracle^{4}$ to $oracle^{128}$ of all settings in the final draft upon acceptance.
>
> > **Q3:** The estimation variance is also critical for this problem, how is the variance of each method (including baselines) under different prompt and settings, does the meta-model shows a small variance across various scenarios? If the authors can elaborate on this point, the method's effectiveness will be further confirmed.
>
> We thank the reviewer for the suggestion. In response, we show the standard deviation of each method across all test prompt permutations below. Notably, in most settings, the meta-models achieve a lower standard deviation than the baseline methods. When compared to the actual accuracy distributions, the standard deviation of the estimation is **significantly smaller**, indicating a non-trivial improvement. Overall the meta-model makes stable accuracy estimations across different settings. We promise to update the variance to the results table upon acceptance.
>
> | Methods | | LLaMA-7B |  |  | LLaMA-13B | |
> | :-----------: | :-----------: | :-----------: | :-----------: | :-----------: |  :-----------: |  :-----------: |
> |  | MMLU  | MCQA | CBQA | MMLU | MCQA | CBQA|
> | MLP | 7.06 &pm; 1.03 | 8.82 &pm; 2.64 | 6.32 &pm; 2.13 |5.80 &pm; 0.63|11.04 &pm; 3.30|6.32 &pm; 2.13|
> |3-NN | 5.98 &pm; 0.97 | 5.62 &pm; 2.35 | 7.10 &pm; 2.15 |5.50 &pm; 0.66| 11.26 &pm; 4.56 | 7.10 &pm; 2.15|
> |XGBoost | 5.42 &pm; 1.09 | 5.22 &pm; 2.17 | 7.00 &pm; 2.44 |5.00 &pm; 0.62| 11.52 &pm; 5.20 | 7.00 &pm; 2.44|
> |AvgTrain| 5.26 &pm; 1.40| 10.26 &pm; 2.50 | 11.62 &pm; 5.88 |9.90 &pm; 1.70| 11.40 &pm; 3.36| 11.62 &pm; 5.88|
> |AvgConf| 5.54 &pm; 0.54| 6.88 &pm; 3.14 |n/a |5.10 &pm; 0.86|8.58 &pm; 1.75|n/a|
> |TS|5.6 &pm; 1.63| 6.32 &pm; 3.58 |n/a|14.06 &pm; 1.53|14.00 &pm; 6.95|n/a|
> |ATC|20.34 &pm; 4.1| 34.66 &pm; 9.36 |31.8 &pm; 13.44|20.50 &pm; 4.92|24.14 &pm; 7.72|31.80 &pm; 13.44|
> |ACC| 31.08 &pm; 6.32 | 39.00 &pm; 10.52 | 23.70 &pm;10.44 | 45.50 &pm; 11.74 | 50.34 &pm; 12.84 | 29.30 &pm; 12.38|
>
> | Methods | | OPT-6.7B |  |  | OPT-13B | |
> | :-----------: | :-----------: | :-----------: | :-----------: | :-----------: |  :-----------: |  :-----------: |
> |  | MMLU  | MCQA | CBQA | MMLU | MCQA | CBQA|
> | MLP | 5.70 &pm; 0.77 | 6.66 &pm; 1.25 | 7.28 &pm; 2.75 | 6.30 &pm; 1.14 | 7.18 &pm; 1.69 | 5.90 &pm; 1.18 |
> |3-NN | 4.06 &pm; 0.33 | 2.98 &pm; 0.57 | 6.46 &pm; 2.05 | 5.16 &pm; 0.24 | 2.78 &pm; 1.06 | 5.84 &pm; 2.19 |
> |XGBoost | 3.76 &pm; 0.32 | 2.60 &pm; 0.73 | 8.06 &pm; 3.07 | 4.54 &pm; 0.35 | 2.54 &pm; 1.16 | 6.00 &pm; 2.43 |
> |AvgTrain| 3.48 &pm; 0.37 | 10.66 &pm; 1.75 | 8.38 &pm; 3.86 | 4.28 &pm; 0.35 | 11.46 &pm; 2.15 | 18.00 &pm; 3.77|
> |AvgConf| 8.42 &pm; 0.60 | 13.42 &pm; 1.70 | n/a | 6.20 &pm; 0.77 | 7.14 &pm; 1.77 | n/a|
> |TS| 3.92 &pm; 0.17 | 4.60 &pm; 0.80 | n/a | 4.36 &pm; 0.31 | 2.72 &pm; 1.03 | n/a|
> |ATC| 24.54 &pm; 2.12 | 37.64 &pm; 7.93 | 32.16 &pm; 14.60 | 24.72 &pm; 3.32 | 29.4 &pm; 10.28 | 30.34 &pm; 12.48|
> |ACC| 26.68 &pm; 4.42 | 33.16 &pm; 9.80 | 17.54 &pm; 5.92 | 26.82 &pm; 5.28 | 33.68 &pm; 12.06 | 19.34 &pm; 6.30|
>
> > **Q4:** Line 78 'conducting extensive ICL experiments spanning 91 datasets', I didn't see the results on 91 datasets in the subsequent sections.
>
> We are sorry for the confusion. 91 is the total number of datasets across all 3 dataset collections (MMLU, MCQA, and CBQA). The full dataset collection is in Table 5 in the appendix. For evaluation results, we show the average results across all the datasets in each dataset collection.
>
> > **Q5:** It seems that the problem is highly influenced by label distribution of a task?
>
> We assume that the reviewer refers to the label space of the multiple-choice QA tasks as the closed-book QA tasks would not have a fixed label space. We would like to clarify that all implementations of the dataset collections are from Huggingface Datasets and there is no sign of label distribution skewness in our experiment data. Meanwhile, there is no evidence that the estimation results would be affected by the label distributions of tasks.
>
> *We hope our responses have addressed the concerns of the reviewer and we are happy to answer any further questions.*

---

### Official Review · Reviewer_rur3 · 2023-08-06

**Soundness:** 3

**Excitement:**

4: Strong: This paper deepens the understanding of some phenomenon or lowers the barriers to an existing research direction.

**Paper Topic And Main Contributions:**

The paper is focusing on the practical value of in-context learning (ICL) with large language models (LLMs). Specifically, the authors argue that, naturally, when evaluating ICL with a labeled test set is a direct solution to know whether ICL will be effective, it greatly reduces the appeal of ICL, as one of ICL’s key selling points is that it does not require a large labeled dataset. In addition, many tasks do not come with a labeled test set due to high annotation costs (e.g., medical/law-related questions that require professional knowledge to answer). The authors thus introduce the problem of few-shot ICL accuracy estimation: given a handful of labeled in-context examples and a set of unlabeled test examples, the goal is to estimate the overall accuracy of ICL on these test examples. They test a series of models (e.g., MLP and XGBoost) to train a meta-model for accuracy estimation. They evaluate the models against baselines and oracle in 3 datasets and 4 models, showing the potential value of the proposed approach.

**Questions For The Authors:**

- In Section 3.1, how the prompt c^~ (L177)  is different than prompt c?
- What does j represent in L188 and consequently what does c_ij refer to? What does K prompt mean? Isn't there a single prompt per task?
- Why do the authors think that the meta models perform better on the CBQA datasets compared to the others? What could be the explanation that there are not consistent results/patterns?

**Reasons To Accept:**

- The paper is well written and easy to follow. The problem studied is well motivated and of practical importance.
- The experimental setup is solid and the experiments thorough (multiple baselines, models, datasets).
- The authors proposed several (existing) methods to constitute the model architecture for training the meta-model, which shows to be performing well in some settings. They also provide an ablation study to explain design choices.

**Reasons To Reject:**

- I am not sure how a meta-model can be useful to solve the problem (of requiring labeled test data) in practice, since labelled data will always be needed to train a meta-model
- Even though the problem formulation (i.e., ICL accuracy estimation) is interesting and novel, the proposed methods to address it are not.
- The results are not convincing that the proposed approach can be actually beneficial in a real low (labaled) data resource setting (e.g., 'In 2 out of 12 settings, they match with evaluating on 128 labeled examples.').

**Reproducibility:**

4: Could mostly reproduce the results, but there may be some variation because of sample variance or minor variations in their interpretation of the protocol or method.

**Reviewer Confidence:**

4: Quite sure. I tried to check the important points carefully. It's unlikely, though conceivable, that I missed something that should affect my ratings.

**Typos Grammar Style And Presentation Improvements:**

- In 5.2 there are several 'orcale' instead of 'oracle'.

---

> ### Author Rebuttal · Authors · 2023-08-29
>
> We thank the reviewer for the insightful questions and comments and we are glad that the reviewer appreciates the paper writing and acknowledges the practicality of our proposed methods. Here are our responses to the individual questions:
>
> > **Q1:** I am not sure how a meta-model can be useful to solve the problem (of requiring labeled test data) in practice, since labeled data will always be needed to train a meta-model
>
> We are sorry for the confusion. The meta-model would be useful for estimating in-context learning accuracy for test datasets without acquiring labeled test data, which is particularly beneficial for the low-resource tasks that annotations are expensive to collect. On the other hand, we are able to generate meta-training data from observations of LLMs performance on datasets with plenty of labeled examples readily available (Sec 4.3, L266). We then train the meta-model using the meta-training data to estimate the performance on the test datasets, therefore saving the cost of collecting annotations for the test datasets. Figure 1 provides a visual demonstration of our task setting.
>
> > **Q2:** Even though the problem formulation (i.e., ICL accuracy estimation) is interesting and novel, the proposed methods to address it are not.
>
> We proposed the confidence metric for free-form generation and proved that the ICL accuracy estimation can be done with a simple meta-model. We are the first to apply such confidence metrics to estimate in-context learning accuracies on a dataset level.
>
> > **Q3:** The results are not convincing that the proposed approach can be actually beneficial in a real low (labaled) data resource setting (e.g., 'In 2 out of 12 settings, they match with evaluating on 128 labeled examples.').
>
> We would like to clarify:
> * The oracle baselines indicate how many labeled test data we could otherwise save from annotating to achieve the same estimation accuracy. A high oracle baseline (e.g.,  $oracle^{128}$) would represent a high estimation accuracy.
> * As mentioned by the reviewer, the meta-model is not able to achieve $oracle^{128}$ in every setting. However, it is able to achieve  $oracle^{16}$ in 10 out of 12 settings, which still represent the **non-trivial** cost of annotation 16 examples for the real low resource setting. Therefore, in the practical case, it is highly likely that the meta-model could be still beneficial to save a substantial amount of expensive annotations.
>
> > **Q4:** In Section 3.1, how the prompt $c^{~}$ (L177) is different than prompt $c$?
>
> We are sorry for the confusion. $c^{~}$ is used to denote the prompts for the labeled datasets (seen datasets) as opposed to the prompts for the unlabeled datasets (unseen test datasets) $c$. A subscript could work better here but might look messy as we incorporate c in the accuracy notations. (e.g., L179)
>
> > **Q5:** What does $j$ represent in L188 and consequently what does $c_{ij}$ refer to? What does $K$ prompt mean? Isn't there a single prompt per task?
>
> We are sorry for the confusion. For each task, we have several prompt templates (sec B in appendix, L765), and each prompt template is followed by in-context example permutations (L418). $c_{ij}$ refers to one single prompt, and $K$ is the total number of prompt variations. For example, there are 5 prompt templates for MMLU datasets. For each dataset, we generate 30 in-context example permutations, which results in $K=5\times30=150$ prompts. Each $c_{ij}$ refers to one prompt in these $K$ prompts.
>
> > **Q6:** Why do the authors think that the meta-models perform better on the CBQA datasets compared to the others? What could be the explanation that there are not consistent results/patterns?
>
> We would like to draw attention to Table 2 in the appendix where we include the full results. To clarify, we did not claim that meta-models perform better on the CBQA datasets. However, the meta-models appear to more frequently outperform the baselines because the baseline methods do not estimate as accurately for free-form generation tasks as closed-set generation tasks. One possible explanation for the inconsistency is that free-form generation tasks have different confidence metrics from closed-set generation tasks (Sec 4.2, L241).
>
> We thank the reviewer for the meticulous review and will fix all the typos.
>
> *We hope our responses have addressed the concerns of the reviewer and we are happy to answer any further questions.*

---

### Official Review · Reviewer_8wF1 · 2023-08-11

**Soundness:** 3

**Excitement:**

3: Ambivalent: It has merits (e.g., it reports state-of-the-art results, the idea is nice), but there are key weaknesses (e.g., it describes incremental work), and it can significantly benefit from another round of revision. However, I won't object to accepting it if my co-reviewers champion it.

**Paper Topic And Main Contributions:**

The paper under review presents a study on estimating in-context learning (ICL) accuracies for both seen and unseen tasks, utilizing minimal ground truth labels. The authors introduce a meta-model approach and evaluate it across three datasets and two language model architectures (OPT and LLaMA), demonstrating its superior performance over baselines. The inclusion of confidence-based evaluation strategies adds further value to the paper, given their relevance to empirical practices in the NLP community.

**Questions For The Authors:**

An intriguing avenue for future research would be extending this estimation technique to encompass other performance metrics like sensitivity, calibration, and selective prediction. Such extensions could significantly enhance the practical utility of the approach, especially concerning scenarios where the model abstains from answering due to low confidence.

Do authors have any opinions on this?

**Reasons To Accept:**

* The paper addresses a significant and practical concern in NLP, namely, the estimation of ICL performance using a meta-model framework. This approach has the potential to considerably save time and resources.

* The comparative evaluation against baselines showcases the paper's effectiveness. The results highlight its state-of-the-art performance, which is a strong indicator of the approach's viability.

* The ablative analysis enhances the clarity of the proposed system. By discerning the impact of different components, the authors provide insights into the workings of the model and its underlying mechanisms.

**Reasons To Reject:**

* The observed improvements appear modest, particularly when juxtaposed with the fact that the Oracle method only requires 32 examples for accurate estimation. Given the relatively manageable nature of manually evaluating 32 examples, the incremental benefits of the proposed approach could be perceived as limited.

* The evaluations are weak. Expanding the evaluation to include a more diverse set of datasets, such as the Natural-Instruction V1/2, would enhance the generalizability of the findings and bolster the argument for the approach's robustness.

* The high variance in accuracy, as evident from Table 1, might diminish the perceived significance of the improvements. Integrating a statistical test for a more rigorous comparison could provide a more compelling case.

* The reporting of the lower-bound $oracle^{32}$ might be revised. Instead, a graphical representation comparing Oracle's accuracy against other methods with an increasing number of examples would offer a more intuitive and informative visualization.

**Reproducibility:**

4: Could mostly reproduce the results, but there may be some variation because of sample variance or minor variations in their interpretation of the protocol or method.

**Reviewer Confidence:**

4: Quite sure. I tried to check the important points carefully. It's unlikely, though conceivable, that I missed something that should affect my ratings.

---

> ### Author Rebuttal · Authors · 2023-08-29
>
> We thank the reviewer for the insightful questions and comments! We are glad that the reviewer found our paper to be practically interesting and useful. Here are our responses to the individual questions:
>
> > **Q1:** The observed improvements appear modest, particularly when juxtaposed with the fact that the Oracle method only requires 32 examples for accurate estimation. Given the relatively manageable nature of manually evaluating 32 examples, the incremental benefits of the proposed approach could be perceived as limited.
>
> We believe that matching the estimation accuracy with collecting 32 labeled examples could be a substantial improvement.
>
> * The annotation cost varies by the domain. For some domains that require professional knowledge to annotate, collecting 32 examples might already be costly.
> * One main feature of our method is the ability to generalize to new test datasets. If a practitioner wanted to estimate accuracy on many out-of-distribution datasets, our method would become more advantageous, since it can save 32 labeled examples per dataset.
>
> Lastly, part of our contribution is to establish the evaluation setting as a benchmark, and we would be happy to see future work improve on our baselines.
>
> > **Q2:** The evaluations are weak. Expanding the evaluation to include a more diverse set of datasets, such as the Natural-Instruction V1/2, would enhance the generalizability of the findings and bolster the argument for the approach's robustness.
>
> We agree with the reviewer that including more datasets would help prove the robustness of the proposed method, but we would like to clarify that we conduct experiments on three collections of Question-Answering datasets containing 91 datasets in total, and there are 4 LLMs that we experimented with. Considering the number of permutations of prompts and in-context examples, we actually evaluate a total of **42360** in-context learning performance observations. While we would be interested in running experiments on additional datasets, the number of datasets we can evaluate is subject to computational resource limitations and we consider exploring such generalizability as future work.
>
> > **Q3:** The high variance in accuracy, as evident from Table 1, might diminish the perceived significance of the improvements. Integrating a statistical test for a more rigorous comparison could provide a more compelling case
>
> We are sorry about the possible confusion. To clarify, the variances reported in the table refer to the variance in in-context learning performance across all datasets and prompt permutations (see Figure 5 in the appendix for an illustration). The high variances indicate that accuracy estimation is nontrivial, as a constant estimator cannot achieve low error, so this actually demonstrates that our calibrator is performing a nontrivial task. For example, in our best case, the variance of actual accuracy is 12.06 whereas the estimation error is just 2.54. Across all 12 settings, the best estimations from the meta-models are 67.5% of the standard deviations, and in 11 out of 12 settings, the estimation error is less than one standard deviation.
>
> We show the accuracy estimation variance across all test prompts for all settings below. It is evident that the meta-model estimation standard deviations are **significantly smaller** than the standard deviation of the actual accuracy, which indicates non-trivial improvements in accuracy estimations.
>
> | Methods | | LLaMA-7B |  |  | LLaMA-13B | |
> | :-----------: | :-----------: | :-----------: | :-----------: | :-----------: |  :-----------: |  :-----------: |
> |  | MMLU  | MCQA | CBQA | MMLU | MCQA | CBQA|
> | MLP | 7.06 &pm; 1.03 | 8.82 &pm; 2.64 | 6.32 &pm; 2.13 |5.80 &pm; 0.63|11.04 &pm; 3.30|6.32 &pm; 2.13|
> |3-NN | 5.98 &pm; 0.97 | 5.62 &pm; 2.35 | 7.10 &pm; 2.15 |5.50 &pm; 0.66| 11.26 &pm; 4.56 | 7.10 &pm; 2.15|
> |XGBoost | 5.42 &pm; 1.09 | 5.22 &pm; 2.17 | 7.00 &pm; 2.44 |5.00 &pm; 0.62| 11.52 &pm; 5.20 | 7.00 &pm; 2.44|
> |AvgTrain| 5.26 &pm; 1.40| 10.26 &pm; 2.50 | 11.62 &pm; 5.88 |9.90 &pm; 1.70| 11.40 &pm; 3.36| 11.62 &pm; 5.88|
> |AvgConf| 5.54 &pm; 0.54| 6.88 &pm; 3.14 |n/a |5.10 &pm; 0.86|8.58 &pm; 1.75|n/a|
> |TS|5.6 &pm; 1.63| 6.32 &pm; 3.58 |n/a|14.06 &pm; 1.53|14.00 &pm; 6.95|n/a|
> |ATC|20.34 &pm; 4.1| 34.66 &pm; 9.36 |31.8 &pm; 13.44|20.50 &pm; 4.92|24.14 &pm; 7.72|31.80 &pm; 13.44|
> |ACC| 31.08 &pm; 6.32 | 39.00 &pm; 10.52 | 23.70 &pm;10.44 | 45.50 &pm; 11.74 | 50.34 &pm; 12.84 | 29.30 &pm; 12.38|
>
> | Methods | | OPT-6.7B |  |  | OPT-13B | |
> | :-----------: | :-----------: | :-----------: | :-----------: | :-----------: |  :-----------: |  :-----------: |
> |  | MMLU  | MCQA | CBQA | MMLU | MCQA | CBQA|
> | MLP | 5.70 &pm; 0.77 | 6.66 &pm; 1.25 | 7.28 &pm; 2.75 | 6.30 &pm; 1.14 | 7.18 &pm; 1.69 | 5.90 &pm; 1.18 |
> |3-NN | 4.06 &pm; 0.33 | 2.98 &pm; 0.57 | 6.46 &pm; 2.05 | 5.16 &pm; 0.24 | 2.78 &pm; 1.06 | 5.84 &pm; 2.19 |
> |XGBoost | 3.76 &pm; 0.32 | 2.60 &pm; 0.73 | 8.06 &pm; 3.07 | 4.54 &pm; 0.35 | 2.54 &pm; 1.16 | 6.00 &pm; 2.43 |
> |AvgTrain| 3.48 &pm; 0.37 | 10.66 &pm; 1.75 | 8.38 &pm; 3.86 | 4.28 &pm; 0.35 | 11.46 &pm; 2.15 | 18.00 &pm; 3.77|
> |AvgConf| 8.42 &pm; 0.60 | 13.42 &pm; 1.70 | n/a | 6.20 &pm; 0.77 | 7.14 &pm; 1.77 | n/a|
> |TS| 3.92 &pm; 0.17 | 4.60 &pm; 0.80 | n/a | 4.36 &pm; 0.31 | 2.72 &pm; 1.03 | n/a|
> |ATC| 24.54 &pm; 2.12 | 37.64 &pm; 7.93 | 32.16 &pm; 14.60 | 24.72 &pm; 3.32 | 29.4 &pm; 10.28 | 30.34 &pm; 12.48|
> |ACC| 26.68 &pm; 4.42 | 33.16 &pm; 9.80 | 17.54 &pm; 5.92 | 26.82 &pm; 5.28 | 33.68 &pm; 12.06 | 19.34 &pm; 6.30|
>
> > **Q4:** The reporting of the lower-bound $oracle^{32}$ might be revised. Instead, a graphical representation comparing Oracle's accuracy against other methods with an increasing number of examples would offer a more intuitive and informative visualization.
>
> We thank the reviewer for the great suggestion. We prepared the graphical representations of the oracle baselines from $oracle^{4}$ to $oracle^{128}$ of all settings but did not include them in the final draft. Since the oracle baselines vary across different settings, it would be messy to concatenate the graphs together. We agree that visualizing the oracle baselines would provide more interpretability, and we will include them upon acceptance.
>
> > **Q5:** An intriguing avenue for future research would be extending this estimation technique to encompass other performance metrics like sensitivity, calibration, and selective prediction. Such extensions could significantly enhance the practical utility of the approach, especially concerning scenarios where the model abstains from answering due to low confidence.
>
> We appreciate the reviewer’s idea of how future research can extend the utility of this work. We would like to point out that besides the accuracy metrics that we directly evaluate, the method is built on model calibration as we incorporated model confidence (Sec. 4, L209) as the meta feature. We also take average calibration error as one of the baseline methods to evaluate the model calibration on a dataset level. We consider it as future work to compare the estimation from the meta-model to this baseline to analyze when the LLMs become overconfident.
>
> We also agree with envisioning selective prediction and abstaining from answering as exciting future directions. Meanwhile, it is also worth investigating whether the same method can be applied to estimate the example-level accuracy as opposed to the dataset-level estimation scenario in this work.
>
> *We hope our responses have addressed the concerns of the reviewer and we are happy to answer any further questions.*

---

### Official Review · Reviewer_zZme · 2023-08-14

**Soundness:** 2

**Excitement:**

2: Mediocre: This paper makes marginal contributions (vs non-contemporaneous work), so I would rather not see it in the conference.

**Paper Topic And Main Contributions:**

This paper investigated  how to estimate the in-context learning capability of large language models in the absence of labeled test data. To address this  challenge, the authors developed a meta-model designed to predict the performance of these large language models, complete with an associated confidence features. Specifically, the authors designed a training dataset wherein the input encapsulates the model's confidence features, and the output represents the performance metrics of the large language models. Concurrently, the authors calibrated the model confidence features in alignment with the disparities observed between closed-set generation and open-ended generation. Empirical studies across three question answering tasks, employing two  large language models,  illustrate that the introduced methodology consistently outperforms  baselines.


**Reasons To Accept:**

1. The proposed method offers the capability to predict the in-context learning performance of expansive language models, circumventing the need for human annotations.
2. It has been empirically demonstrated that the proposed model consistently outperforms a range of baseline approaches,  based on two distinct large language models.

**Reasons To Reject:**

1. The assessment criteria for the performance of large language models is limited to accuracy metrics. Such a limited view does not necessarily provide a comprehensive representation of the performance of large language models in real-world applications.
2. The method exhibits dependence on similar examples from the training dataset. This raises potential concerns regarding the distribution consistency between the training and test datasets adopted in the study. An in-depth visualization and analysis of the data distributions might be beneficial to address such concerns.
3. The evaluative framework appears somewhat limited in scope. With considerations restricted to merely three Question-Answering tasks and two language models, there are reservations about the method's broader applicability. Its potential to generalize to other reasoning or generation tasks or more advanced models, such as vicunna or alpaca, remains a subject of inquiry.

**Reproducibility:**

3: Could reproduce the results with some difficulty. The settings of parameters are underspecified or subjectively determined; the training/evaluation data are not widely available.

**Reviewer Confidence:**

4: Quite sure. I tried to check the important points carefully. It's unlikely, though conceivable, that I missed something that should affect my ratings.

---

> ### Author Rebuttal · Authors · 2023-08-29
>
> We thank the reviewer for helpful comments and insightful questions! Here are our responses to the questions and concerns:
>
> > **Q1:** The assessment criteria for the performance of large language models is limited to accuracy metrics, which is not a comprehensive representation of the performance of large language models in real-world applications.
>
> We acknowledge that there is more than one dimension to evaluate large language model capabilities as outlined in [1], but in this work, our main focus is to estimate in-context learning accuracies for QA tasks, for which the accuracy metrics are particularly important. Meanwhile, we would like to point out that the method we use to estimate accuracy also evaluates model calibration and robustness, as we incorporated model confidence (Sec. 4, L209) as the meta feature, and estimate performance on datasets that are out-of-domain (L404).
>
> > **Q2:** The method exhibits dependence on similar examples from the training dataset. This raises potential concerns regarding the distribution consistency between the training and test datasets adopted in the study. An in-depth visualization and analysis of the data distributions might be beneficial to address such concerns.
>
> We hope to clarify that the test datasets are out-of-distribution from the training datasets.
>
> * There exists **domain shifts** between datasets within the same collection (Table 5 in the appendix). For example, the *abstract algebra* dataset would be an out-of-distribution dataset from the *anatomy* dataset in MMLU.
> * We perform 5-fold cross-validation for each dataset collection, which prevents overfitting to possible training/test consistency patterns. Except that the meta-training/test data share the same task format (e.g., multiple-choice QA), we would not consider them as any more similar.
>
> For further clarification, we would like to point out that the reason we choose not to test the cross-task generalizability (e.g., from multiple-choice QA datasets to closed-book QA datasets) is that they have different accuracy metrics (Exact match v.s. F1-score). Although it is possible to solve multiple-choice QA tasks as a generation task and report F1-score instead, it is not the common practice.
>
> We promise to include a visualization of domain distributions for the dataset collections upon acceptance.
>
> > **Q3:** The evaluative framework appears somewhat limited in scope. With considerations restricted to merely three Question-answer tasks and two language models, there are reservations about the method's broader applicability. Its potential to generalize to other reasoning or generation tasks or more advanced models, such as vicuna or alpaca, remains a subject of inquiry.
>
> We conduct experiments on three collections of Question-Answering datasets containing **91** datasets in total, and there are **4** LLMs that we experimented with. Considering the number of permutations of prompts and in-context examples, we actually evaluate a total of **42360** in-context learning performance observations. While we would be interested in running experiments on additional datasets, the number of datasets we can evaluate is subject to computational resource limitations and we consider exploring such generalizability as future work.
>
> We acknowledge that instruction-tuned models are of great interest to the community recently, but in our task setting, some concerns need to be addressed.
>
> * We aim to estimate in-context learning performance and the LLM of interest should therefore be capable of performing in-context learning, but many instruction-tuned models such as Vicuna and Alpaca are not instruction-tuned to do few-shot in-context learning.
> * Instruction-tuning sometimes hurts model performance on canonical datasets such as MMLU, as shown in [2], and might also significantly hurt calibration as reported in [3].
> * To evaluate an additional language model we would need to evaluate around 10,000 in-context learning settings. We therefore did not include instruction-tuned models in the study, but consider it future work to investigate whether the same framework applies to such models.
> * We do not consider LLMs that are directly trained on our evaluation datasets such as T0 to prevent data leakage.
>
> *We hope our responses have addressed the concerns of the reviewer and we are happy to answer any further questions.*
>
> References:
>
> [1] Liang, Percy, et al. "Holistic evaluation of language models." arXiv preprint arXiv:2211.09110 (2022)
>
> [2] Gudibande, Arnav, et al. "The false promise of imitating proprietary llms." arXiv preprint arXiv:2305.15717 (2023).
>
> [3] OpenAI. “GPT 4 Technical Report”. arXiv preprint arXiv:2303.08774 (2023).

---

### Meta-Review · Area_Chair_jmHC · 2023-09-19

**Recommendation:** 3

**Metareview:**

This paper presents a meta-model to estimate the accuracy and confidence of other language models with in-context learning in the absence of labeled test data. This approach has the potential to considerably save time and resources, and the experiments are solid and well-motivated. Multiple reviewers expressed concerns about the applicability of this method, but this issue seems to have been answered in the rebuttals.

---

### Decision · Program_Chairs · 2023-10-07

**Decision:**

Accept-Findings

**Comment:**

This paper presents a meta-model to estimate the accuracy and confidence of other language models with in-context learning in the absence of labeled test data. This approach has the potential to considerably save time and resources, and the experiments are solid and well-motivated. Multiple reviewers expressed concerns about the applicability of this method, but this issue seems to have been answered in the rebuttals.